# Comparative Study on Distributed Lightweight Deep Learning Models for Road Pothole Detection

**DOI:** 10.3390/s23094347

**Published:** 2023-04-27

**Authors:** Hassam Tahir, Eun-Sung Jung

**Affiliations:** Department of Software & Communications Engineering, Hongik University, Sejong 30016, Republic of Korea; hassam_tahir@g.hongik.ac.kr

**Keywords:** distributed deep-learning, distributed edge AI/ML, distributed hybrid model training

## Abstract

This paper delves into image detection based on distributed deep-learning techniques for intelligent traffic systems or self-driving cars. The accuracy and precision of neural networks deployed on edge devices (e.g., CCTV (closed-circuit television) for road surveillance) with small datasets may be compromised, leading to the misjudgment of targets. To address this challenge, TensorFlow and PyTorch were used to initialize various distributed model parallel and data parallel techniques. Despite the success of these techniques, communication constraints were observed along with certain speed issues. As a result, a hybrid pipeline was proposed, combining both dataset and model distribution through an all-reduced algorithm and NVlinks to prevent miscommunication among gradients. The proposed approach was tested on both an edge cluster and Google cluster environment, demonstrating superior performance compared to other test settings, with the quality of the bounding box detection system meeting expectations with increased reliability. Performance metrics, including total training time, images/second, cross-entropy loss, and total loss against the number of the epoch, were evaluated, revealing a robust competition between TensorFlow and PyTorch. The PyTorch environment’s hybrid pipeline outperformed other test settings.

## 1. Introduction

A self-driving automobile employs an artificial intelligence (AI) system to evaluate data from sensors and make judgments while driving [1]. The disposition of smart cars assails as a fast catalyst for the revolutionary steps toward the future of intelligent transportation systems by decreasing pollution, reducing accidents, and decreasing traffic [2]. The self-driving vehicles will remarkably decrease road accidents in the future through human input integrated with AI programs. For self-driving cars, crack detection is crucial because these vehicles rely on sensors to perceive and navigate the environment. If cracks are not detected and repaired promptly, they can interfere with the vehicle’s perception systems, leading to incorrect or incomplete information about the road ahead. Crack visualization has certain methods, such as the use of a deep-learning architecture, capable of processing images at multiple scales [3] and detecting strains in columns via the mark-free vision methodology [4]. However, substantial doubts about reliability, regulations, and predictive detection have been encountered and raised [5]. Most reported accidents of self-driving cars were due to inappropriate or heavy-weight neural network training on edge devices, resulting in heating issues [6]. The inability to judge the difference between potholes or patches results in the sudden break or non-breaking elements at inappropriate places because of a confused state of the neural network [7]. In this regard, detecting potholes in self-driving vehicles or road maintenance is vital for future intelligent transportation systems. The requirements of pothole-detecting AI systems include the following: (1) a lightweight distributed neural network, (2) high-quality input images for training in a distributed edge cloud environment, and (3) reliable communication for appropriate information exchange among distributed deep learning [8].

Over the past few years, distributed deep learning has emerged as a promising area of research, supported by scalable cutting-edge technology and driven by the need to tackle large-scale datasets and complex problems. Numerous state-of-the-art studies have been conducted to develop innovative techniques and frameworks for optimizing distributed deep-learning systems. For instance, exploring data and model parallelism has led to significant advancements in the scalability and efficiency of training large neural networks [9]. Additionally, researchers have been investigating the impact of communication strategies, such as gradient compression [10] and decentralized optimization [11], to reduce the communication overhead and latency associated with the distributed training process. Furthermore, novel approaches, such as federated learning [12], have been proposed to enable collaborative learning among multiple devices while preserving data privacy. These studies reflect ongoing efforts to develop more efficient, scalable, and privacy-preserving distributed deep-learning systems, ultimately contributing to the broader applicability of deep learning in various domains.

There are two main distributed learning strategies. The first strategy is data parallelism [13]. An extensive dataset is common for more accurate results in modern deep learning. Due to the extensive dataset, the memory fitting problem occurs vastly. To overcome this issue, the large dataset is divided into small batches, and their gradients are calculated individually on different GPUs; the final result is the weighted average of all the calculated gradients. Furthermore, the second technique is model parallelism [14]. Model parallelism is required when the model (layers of the model) or parameters are too large to fit in the memory. Therefore, deep-learning models could be divided into pieces; a few consecutive layers could be transferred to a single node, and the gradients could be calculated in the forward direction. Synchronous [15] and asynchronous training [16] is a typical method to solve data/model parallelism.

Majorly, two main libraries support distributed learning: TensorFlow and PyTorch. TensorFlow is used vastly in industrial products and provides distributed APIs for data distribution across multiple devices (e.g., GPU and TPU). Users can distribute data and create a training pipeline with minimal changes in the code. One of the significant drawbacks of TensorFlow’s distributed APIs is that they support model distribution but with many limitations. On the other hand, PyTorch’s distributed APIs are fastly growing in model distribution and data distribution [17]. PyTorch contains various model and data parallelism options according to the user’s requirements [18]. PyTorch also provides flexibility to develop its model and data distribution training pipeline.

However, the increase in computational capabilities is significantly outpaced by the expansion of the datasets and models [19], which is why, even after achieving the distribution scenarios of training, the production deployment of these networks remains premature [20]. Consequently, the memory capacity and communication overhead can limit the scaling of data parallelism.

### 1.1. Our Technical Contributions

In this study, we developed and presented a performance comparison of data and model distribution on an edge cluster testbed by varying the number of graphics processing units (GPUs). The edge cluster testbed was designed to emulate resource-constrained environments and integrated several GPUs with Kubernetes services for seamless container orchestration. This setup facilitated the distribution of diverse workloads across end-to-end ML/AI workflows, eliminating concerns about underlying memory issues in resource-constrained scenarios.

To further enhance the efficiency of the edge cluster testbed, we proposed a hybrid pipeline that combined data parallelism and model parallelism for the simultaneous training of machine learning models. By concurrently dividing model layers and data shards across the available compute resources, this approach exploited the advantages of both parallelism techniques. This hybrid pipeline optimized resource utilization in edge computing environments, leading to accelerated training times and improved scalability for deep-learning workloads.

In our performance analysis, we employed a meticulous experimental design that incorporated a representative deep-learning model (convolutional neural networks (CNNs)), which is widely used in self-driving car applications. We utilized relevant datasets, ensuring that they accurately reflected real-world scenarios encountered by autonomous vehicles. Key performance metrics, including training time and images per second, model accuracy, and model loss, were rigorously evaluated to provide a comprehensive understanding of each approach’s efficacy.

The comparison of the hybrid distributed pipeline with established frameworks, such as PyTorch and TensorFlow APIs, was performed on both edge devices and the Google Cloud Platform. This cross-platform evaluation facilitated a thorough investigation of the strengths and limitations of each approach in diverse computing environments, thereby enabling us to draw robust conclusions about their suitability for real-time analysis in self-driving car applications.

By employing state-of-the-art techniques and rigorous experimental design, our study showcases the potential of the hybrid distributed pipeline in addressing the unique challenges of edge computing scenarios, particularly in the context of autonomous vehicles. Our comprehensive performance evaluation not only demonstrates the technical superiority of the hybrid approach but also establishes it as a viable and efficient solution for enabling real-time deep-learning processing, ultimately contributing to the advancement of self-driving car technology.

Keeping in view the technical limitations of the current methods mentioned in the next section, we proposed a hybrid pipeline for distributed training and showed the efficacy of our approach through comparative performance analysis on the Google Cloud Platform and our edge cluster testbed. The edge cluster testbed comprised three GPUs, and we used TensorFlow and PyTorch distributed API for implementation. More concrete contributions are as follows:We showed the performance comparison of data and model distribution on the edge cluster testbed by varying the number of GPUs. The edge cluster testbed integrated several devices (GPUs) with Kubernetes services. The diverse workloads are easily distributed by end-to-end ML/AI workflows using Kubernetes services without worrying about underlying memory issues in resource-constricted environments;We formulated a hybrid (data and parallel) pipeline to simultaneously train the model by simultaneously dividing model layers and data shards on the edge cluster;We conducted a performance comparison of the proposed hybrid distributed pipeline employing data and model distribution simultaneously on edge clusters for up to four GPUs. We also compared the proposed hybrid approach vs. PyTorch APIs vs. TensorFlow APIs on both edge devices and the Google Cloud Platform to determine the feasibility of proficient real-time analysis for self-driving cars.

The motivation behind the above objectives was to develop and evaluate a highly efficient, scalable, and resource-conscious distributed training approach for deep-learning workloads in edge computing environments. By exploring the performance of data and model distribution with varying numbers of GPUs, designing a hybrid pipeline that combines the strengths of data and model parallelism, and comparing the proposed pipeline against established frameworks on different platforms, our study aimed to identify the most effective solution for real-time analysis in resource-constrained scenarios, such as those encountered in self-driving cars. The successful implementation and evaluation of this approach have the potential to significantly enhance the capabilities of autonomous vehicles, thereby contributing to the broader advancement of deep-learning applications in edge-computing contexts.

### 1.2. Organization of Paper

The paper is organized as follows: Section 2 discusses related work, research gaps, and our contributions; Section 3 presents a comparison of TensorFlow and PyTorch distributed APIs; The proposed hybrid distributed pipeline is detailed in Section 4; The experimental data are provided in Section 5, while Section 6 outlines the experimental settings; The results and discussions are covered in Section 7; and this study’s conclusion is presented in Section 8.

## 2. Related Work

The detection of road potholes using computer vision and machine learning approaches can be a valuable tool to assist with visual challenges [21]. Potholes can pose a significant risk to autonomous vehicles, potentially causing damage to their sensors or suspensions and can lead to accidents or disruptions in traffic flow. Similarly, the automatic detection of pothole distress in asphalt pavement using improved convolutional neural networks (CNNs) is a promising approach for identifying and addressing potholes on time. Potholes can cause significant damage to vehicles, disrupt traffic flow, and pose safety hazards to drivers and pedestrians alike [22].

Similarly, rethinking road surface 3D reconstruction and pothole detection from perspective transformation to disparity map segmentation is a novel approach to detecting and addressing potholes on the road. The traditional method of pothole detection involves using cameras to capture images of the road surface, followed by perspective transformation to create a 3D surface model. However, this method can be time-consuming and computationally expensive [23].

The system known as 3Pod is a federated learning-based system for 3D pothole detection in smart transportation. The system uses a distributed approach where data is collected from various sensors installed in the vehicles and then sent to a centralized server for processing using federated learning techniques. This approach helps improve the accuracy and efficiency of pothole detection while ensuring data privacy. One drawback of this system is that it requires a large amount of computational power and data storage to process and store the 3D point clouds [24].

Traditional distributed deep-learning pothole detection systems may not be accurate or reliable enough for use in self-driving cars, as they may be affected by various factors, such as lighting conditions, weather, and road surface variations. Moreover, the system’s reliability is dependent on both hardware and software. The system’s hardware components, such as the sensors and processors, must be able to accurately capture and process data for the software to analyze and interpret it effectively. Therefore, it is essential to ensure that the hardware is high-quality and meets the necessary specifications. To achieve cutting-edge development, high-end distributed strategies should be developed. Developing a high-end distributed environment for pothole and road distress detection, as a use case of self-driving cars, requires an in-depth understanding of distributed deep learning.

The distributed model analysis is thought to be the foundation of an Oracle tool that can help to identify limitations and bottlenecks of various parallelism approaches during their scaling scenario. This methodology assesses Oracle using six parallelization algorithms, four CNN models, and different datasets (2D and 3D) on up to 1024 GPUs. Compared to empirical results, the Oracle tool has an average accuracy of roughly 86.74% and data parallelism accuracy of up to 97.57% [25]. However, GPU processing performance and training throughput are severely limited because of the excessive memory consumption mentioned before.

To tackle the issue mentioned above, a model named Hippie was proposed [26]. Hippie is a hybrid parallel training framework that combines pipeline and data parallelism to increase the memory economy and scalability of massive DNN training. Hippie uses a hybrid parallel approach based on hiding gradient communication to boost training throughput and scalability. Hippie was created utilizing the PyTorch and NCCL platforms. According to tests on diverse models, Hippie achieved above 90% scaling efficiency on a 16-GPU architecture. Hippie also boosts performance by up to 80% while reducing memory overhead by 57%, resulting in a memory efficiency of 4.18×. However, significant speed-up issues were observed in inherently sequential tasks.

HyPar-Flow is a single API for processing data, model, and hybrid parallel training at scale for any Keras model. To accumulate/average gradients across model replicates, the all-reduce algorithm is employed. HyPar-Flow presents a significant advancement in distributed training, as it provides several notable benefits. First, it offers up to 1.6 times the speed of Horovod-based (Horovod is an open-source package that overcomes both scaling challenges in inter-GPU communication [27]) data-parallel training in sequential tasks, demonstrating its superior efficiency. Second, HyPar-Flow can achieve 110 times the speed of a single node when deployed, showcasing its impressive scalability. Lastly, for ResNet-1001, an ultra-deep model, HyPar-Flow boasts an astounding 481 times the speed of single-node performance when implemented on 512 Frontera nodes, further emphasizing its remarkable capabilities in handling complex and resource-intensive tasks. While the aforementioned information highlights the impressive performance and scalability of HyPar-Flow, it does not address the potential increase in communication overhead due to the combination of data and model parallelism in HyPar-Flow, which could impact its overall efficiency in specific scenarios.

Communication overhead is one of the most significant roadblocks to training big deep-learning models at scale. Gradient sparsification is a promising technique for reducing the amount of data transmitted. First, developing a scalable and efficient sparse all-reduce method has proven to be complex. Secondly, the sparsification overhead is crucial in limiting the potential for speed improvement.

The aforementioned issues were addressed for big and distributed deep-learning models by Ok-TOPK, a distributed training system with sparse gradients [28]. Ok-TOPK combines a decentralized parallel stochastic gradient descent (SGD) optimizer with a unique sparse all-reduce technique (less than 6k communication volume and asymptotically optimal). Ok-TOPK achieves model accuracy comparable to dense all-reduce, according to empirical results. Ok-TOPK is more scalable and boosts training performance significantly compared to the optimized dense and state-of-the-art sparse all-reduces (e.g., 3.29×–12.95× improvement for BERT on 256 GPUs).

Furthermore, a distributed framework was introduced for air quality prediction featuring Busan, Republic of Korea as its model city. To forecast the intensity of particle pollution, a deep-learning model was trained on a distributed system known as data parallelism (PM2.5 and PM10) [29]. To determine how the air quality particles are connected in space and time with the dataset distribution, multiple one-dimensional CNN layers are combined with a stacked attention-based BiLSTM layer to extract local spatial features.

The hybrid approach observed in the mentioned research involved asynchronously distributing data and the model within the same algorithm. For instance, the data was initially distributed and trained with the undistributed model, followed by distributing the undistributed model and training it with undistributed data. Additionally, the communication overhead between the GPUs was a more significant concern than the training and epoch time. The research also lacked practical comparisons, as the developed algorithms’ training times were analyzed but not compared to state-of-the-art APIs.

The current literature on distributed training techniques primarily focuses on the performance of these approaches on conventional GPUs, leaving a research gap in understanding their behavior on edge devices, which are critical for real-time applications, such as self-driving cars. To address this gap and enhance the technical value of our research, we proposed a hybrid deep-learning approach tailored for self-driving car applications, specifically for pothole detection on edge devices, such as the Jetson Nano. This approach leveraged sophisticated distributed PyTorch APIs, enabling us to investigate the effectiveness and adaptability of the hybrid pipeline in resource-constrained, real-world scenarios.

## 3. Comparison of TensorFlow and PyTorch Distributed APIs and Their Limitations

TensorFlow and PyTorch are two of the most popular deep-learning frameworks, each offering distributed APIs to facilitate training models across multiple devices, such as GPUs or TPUs. We examined the key differences between the TensorFlow and PyTorch distributed APIs in terms of their architecture, design principles, and functionality.

Architecture:
TensorFlow: TensorFlow employs a dataflow graph-based architecture, where computations are represented as directed acyclic graphs (DAGs). The nodes in the graph represent operations, while the edges represent tensors flowing between these operations. TensorFlow’s distributed training relies on the tf.distribute API, which provides a flexible and extensible way to distribute the training across multiple devices and platforms.PyTorch: PyTorch follows an imperative (eager execution) programming paradigm, which enables dynamic computation graph construction. PyTorch’s distributed training is facilitated by the torch.distributed package, which provides a rich set of communication primitives and backend-specific implementations for distributed training.Design principles:
TensorFlow: The tf.distribute API is designed to be highly modular and easy to use. It offers several distribution strategies (such as MirroredStrategy, MultiWorkerMirroredStrategy, and ParameterServerStrategy) that can be easily incorporated into the existing TensorFlow code with minimal changes.PyTorch: The distributed package aims to provide a simple, flexible, and efficient way to parallelize training across multiple devices. PyTorch offers multiple backends (such as NCCL, Gloo, and MPI) for communication, and supports various parallelization methods, including data parallelism, model parallelism, and hybrid approaches.Functionality:
TensorFlow: The tf.distribute API offers a comprehensive set of features, such as synchronous and asynchronous training, custom training loops, fault tolerance, and checkpointing. It also supports TensorFlow Extended (TFX) components, enabling seamless integration with end-to-end ML pipelines.PyTorch: The distributed package provides various distributed communication primitives, such as all_reduce, scatter, and gather, as well as higher-level abstractions, such as DistributedDataParallel and DistributedSampler. PyTorch’s distributed API also supports advanced features, such as torchelastic, for fault tolerance and elasticity in distributed training.

TensorFlow and PyTorch distributed APIs, despite their powerful distributed training capabilities, have limitations related to static graphs, overhead, manual parallelism, and heterogeneous hardware support. These limitations can affect the efficiency and performance of training in resource-constrained environments, such as edge computing scenarios. A hybrid distribution approach, combining data and model parallelism, addresses these limitations by leveraging the strengths of both techniques, resulting in better resource utilization, improved scalability, and more efficient training in diverse edge computing contexts. As depicted in Figure 1 and Figure 2, the limitations of TensorFlow and PyTorch are illustrated through a comparative analysis. The figures display the impact of these limitations on various significant parameters, with the y-axis representing the severity level ranging from 1 to 5 and the x-axis enumerating the different parameters that exhibit notable constraints. Moreover, Table 1 shows a comparative analysis using values of key features.

## 4. Proposed Hybrid Distributed Pipeline

Scalability is a crucial factor in the development and deployment of self-driving vehicles, as these vehicles generate vast amounts of data and require complex deep-learning models to process this data in real-time. While traditional methods, such as Hadoop or Apache Spark, can address some of the issues related to parallel data processing, these methods can introduce significant overhead during data transfer and may not be sufficient for processing complex deep-learning models.

To address these challenges, this methodology proposes the development of a hybrid distributed pipeline that can efficiently process complex models with massive data in resource-constrained environments while minimizing overhead. The proposed methodology incorporates Kubernetes as the primary source of scalability in the testbed for self-driving vehicles.

The proposed strategy enables efficient resource optimization, load balancing, and effective fault tolerance systems to ensure passenger safety while also providing efficient scaling in response to changing traffic conditions. The integration of the proposed pipeline with Kubernetes-powered testbeds enables self-driving vehicles to scale massively and run efficiently on edge devices.

### 4.1. Deep-Learning Model Architecture

The architecture of FactorNet, as described in [30], consists of four initial layers that take as the input an image with dimensions of 1920 × 1080 and three channels of RGB. Segment 1-a is made up of one convolutional layer with a 7 × 7 filter size and one pooling layer with a 3 × 3 filter. Segment 2-a consists of three layers: the first two layers are convolutional, with filter sizes of 1 × 1 and 3 × 3, and the last layer is a 3 × 3 pooling filter. The architecture incorporates a parallel CNN approach that utilizes four parallel sub-factors. These sub-factors consist of a pooling layer, 3 × 3 convolution, 5 × 5 convolution, and 7 × 7 convolution, respectively. Each sub-factor encodes different visual features and helps to detect object boundaries. The FactorNet architecture is used as the backbone of Faster R-CNN to identify multiple objects in a single frame. The architecture is embedded in the Faster R-CNN framework with the region proposal network (RPN) to reduce visual features by gradually processing them in a parallel manner. The RPN was chosen for its state-of-the-art detection, classification, and prediction capabilities. The architecture of the model can be visualized in Figure 3, as presented in [30]. A significant change in our architecture is the addition of a state-of-the-art Faster R-CNN as the detection front end and the use of FactorNet as the backbone to reduce neural network complexity. The purpose of combining these two models is to assess the computational handling capacity of the proposed pipeline and testbed.

### 4.2. Proposed Hybrid Pipeline

A hybrid distributed pipeline is proposed as shown in Figure 4, which executes model and data distribution simultaneously in the same code using PyTorch API. Initially, the dataset and convolutional neural network model settings are prepared. After that, a training job manager is established, which contains hybrid code with the distribution of the dataset and model incorporated in Kubernetes clusters to initiate the training process on the edge cluster. Furthermore, the linkage of worker nodes is established using integrated all-reduce and NV links to overcome communication overheads. A complete deep neural model is distributed and replicated as a variable in the form of model layers on different devices, denoted as PEs. The dataset is divided into sub-datasets for each model replica in data-parallel. Forward and backward propagation is computed with the help of a micro-batch utilizing different sectors of the dataset. To aggregate the weights, the gradient exchange phase is enabled with the help of the all-reduce algorithm, as shown in Figure 5.

In Figure 4, the hybrid algorithm is explained clearly in the following Algorithm 1. The input dataset is first partitioned into smaller chunks, which are then processed concurrently on different devices. Simultaneously, the model’s layers or subnetworks are distributed across these devices. During the forward and backward passes, the devices exchange intermediate data and gradients to update their respective parts of the model. By combining data parallelism and model parallelism, the algorithm can efficiently utilize multiple devices, thus reducing the overall training time, balancing the computational load, and allowing for the training of larger models that may not fit into a single device’s memory.
**Algorithm 1** Proposed Hybrid Distributed Deep-Learning Detection Algorithm1: **procedure** HybridDistributedTraining
2:        Initialize the distributed environment
3:        Set up data and model parallelism parameters4:        Load the dataset and create distributed data samplers5:        Define the model architecture with the chosen backbone6:        Distribute the model across multiple devices7:        Initialize the optimizer and learning rate scheduler8:        **for** each epoch **do**9:              **for** each batch in training dataset **do**10:                  Distribute data to available devices11:                  Perform data augmentation if necessary12:                  Forward pass through the model13:                  Calculate the loss function14:                  Backward pass to compute gradients15:                  Synchronize gradients across devices16:                  Update the model weights using the optimizer17:                  Update learning rate scheduler if needed18:              **end for**19:              Evaluate the model on the validation dataset20:              **for** each batch in validation dataset **do**21:                  Distribute data to available devices22:                  Forward pass through the model23:                  Calculate the loss function and performance metrics24:              **end for**25:              Calculate the average validation loss and metrics26:              Save the model if the validation score improves27:              Early stopping if validation score does not improve for a set number of epochs28:        **end for**29:        Return the best model30: **end procedure**

In this proposed approach, we defined a neural network model and a training function that takes the rank and world size as arguments. We initialized he distributed training using the dist.init_process_group function and set the device for the current process using torch.cuda.set_device. We then created the model and optimizer and wrapped the model in a DistributedDataParallel object using the DDP constructor. We also created a data loader for the training data. Within the training loop, we moved the data and target to the current device using data.cuda(rank) and target.cuda(rank), and computed the forward pass through the model.

FactorNet f(x) is trained on dataset D for the detection of potholes. The model consisted of n layers, and each layer is divided into m sub-layers. Let $f_{i,j}\left(x\right)$ represent the *j*-th sub-layer of the *i*-th layer of the model. Then, we divided the model into m sub-models, each of which consists of n sub-layers, such that:(1)$$f_{m}\left(x\right)=f\_n,m\left(f\_n,m-1\left(\ldots f\_n,1\left(f\_n-1,m\left(f\_n-1,m-1\left(\ldots f\_1,1\left(x\right)\right)\right)\right)\right)\right)$$

Each sub-model is executed in parallel across different processing units using model parallelism. Let P be the number of processing units.

To train the model, we used the stochastic gradient descent (SGD) algorithm with data parallelism. We divided the dataset *D* into *P* smaller sub-datasets $D_{1},D_{2},\ldots ,D_{P}$, and assigned each sub-dataset to a processing unit. Each sub-dataset contained a subset of the training data.

The notation $f_{m}\left(x;\theta \right)$ represented the sub-model $f_{m}\left(x\right)$ with the parameters θ. Then, the objective function for training the model is as follows:(2)$$\min_{\theta }\frac{1}{\left|D\right|}\sum_{i=1}^{\left|D\right|}L\left(f_{m}\left(x_{i};\theta \right),y_{i}\right)$$
where *L* is the loss function, $x_{i}$ is the input data in batch *i*, and $y_{i}$ is the corresponding target output.

To perform SGD with data parallelism, each processing unit computes the gradients of the objective function with respect to the model parameters θ using their respective sub-dataset. Let $\nabla J(\theta_{i}^{\left(t\right)})$ represent the gradient of the objective function with respect to θ.

Then, the gradients computed by each processing unit are combined using the all-reduce algorithm to update the model parameters. The all-reduce algorithm computes the sum of the gradients across all the processing units and then broadcasts the result back to each processing unit. Moreover, each processing unit *i* computes the gradient of the objective function with respect to θ using their sub-dataset:(3)$$\nabla J\left(\theta_{i}^{\left(t\right)}\right)=\frac{1}{\left|B_{i}\right|}\sum \left(\nabla L(f_{m}\left(x;\theta_{i}^{\left(t\right)}\right),y)\right)$$

Each processing unit *i* sends its gradient to all the other processing nodes using the all-reduce communication protocol and broadcasts the result back to each node:(4)$$\nabla J_{global}=\mathrm{MPI}\_\mathrm{Allreduce}\left(\nabla J\left(\theta_{i}^{\left(t\right)}\right),\mathrm{op}=\;\mathrm{MPI}\_\mathrm{SUM}\right)$$

Furthermore, it computes the average gradient on each node:(5)$$\nabla J_{avg}=\frac{1}{k}\cdot \nabla J_{global}$$

Similarly, the sum of the gradients across all the processing units is computed:(6)$$\nabla J_{\mathrm{global}}=\sum_{i=1}^{k}\nabla J\left(\theta_{i}^{\left(t\right)}\right)$$

The local model parameters θ are updated on node n_i using the computed average gradient:(7)$$\theta_{i}^{\left(t+1\right)}=\theta_{i}^{\left(t\right)}-\alpha \cdot \nabla J_{avg}$$
where α is the learning rate.

The all-reduce algorithm synchronizes gradient updates among multiple workers during deep-learning model training. With four workers (0, 1, 2, and 3), each computes gradients locally, then exchanges and sums them hierarchically in a sequence. After worker 3 receives the sum of all the gradients, it sends the sum back in reverse order, ultimately synchronizing and updating all workers’ models consistently for efficient distributed training, as shown in Figure 5.

As illustrated in Figure 6, the Faster R-CNN model backed with FactorNet is divided into two shards, and the input batch is partitioned into various splits and pipelined into the two model shards. The distinction is that instead of using CUDA streams to parallelize the execution, this research uses asynchronous RPCs. The model shards are stitched together into a single module. In addition, two rpc.remote functions are used to call the two shards on two RPC workers so they can be accessed in the forward pass.

**Figure 6 sensors-23-04347-f006:**
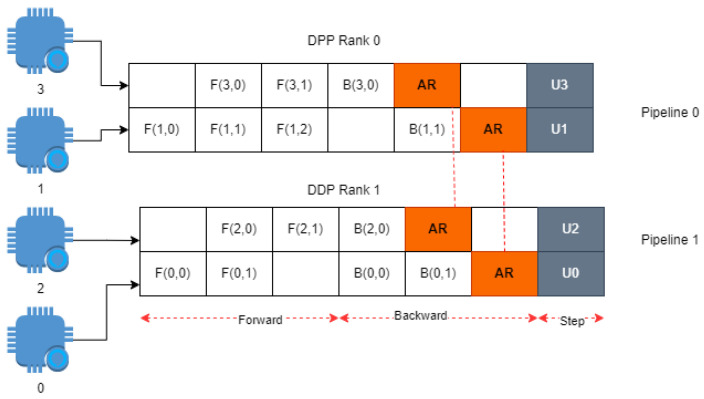
Distribution of FactorNet among hybrid pipeline. Conceptualization from [31,32].

By using hybrid parallelism with the all-reduce algorithm and data awareness, we achieved faster training and better utilization of computing resources. However, the communication overhead, synchronization, and load balancing need to be carefully managed to avoid performance bottlenecks. To avoid this issue in the proposed architecture, NV-Links are introduced via P100 with the all-reduced methodology for gradient information exchange in forward and backward propagation to avoid communication overheads. NV-Links are built with an added feature that enables the overall reduction algorithm to fasten the pipeline and flow gradients in a backup manner. An NV-Link is a direct GPU-to-GPU link that allows the server to scale multi-GPU input/output (IO). Within a single node and between nodes, NV-Switch connects numerous NV-Links to offer all-to-all GPU communication at a complete NV-Link speed [33].

## 5. Experimental Data

The experimental data is divided into two distinct categories. To evaluate the robustness of the hybrid approach under varying conditions, multiple subsets of data have been generated. These subsets serve as diverse representations of the data, providing a more comprehensive examination of the approach’s performance.

To further strengthen the reliability of the proposed approach, these datasets are fed into the model using repetitions of 1× dataset, 4× dataset, and 8× dataset. This is completed in the form of distributed data, which allows the model to adapt and learn from a broader range of input, ultimately enhancing the robustness and dependability of the hybrid approach.

### 5.1. Dataset of Potholes

The dataset used in this study consisted of 10,000 images of potholes captured from various angles as shown in Figure 7. The dataset comprised a combination of local data and various online resources, primarily focusing on road potholes from the USA, Europe, and Southeast Asia. This diverse collection aims to improve the model’s adaptability, a critical factor for self-driving cars. The input size for the pothole images is 1920 × 1080 × 3. To assess the processing speed, the dataset was augmented by replicating the dataset, and different techniques, such as the data parallel, model parallel, and hybrid approaches, were employed. The images were annotated using labelImg software in PASCAL format, and their size was adjusted to meet the neural network’s input requirements. To evaluate the detection of potholes from various dimensions, the images were divided into 70% test data and 30% train data.

### 5.2. Dataset of Road Distress Conditions

The purpose of detecting different road distress conditions is to train a distributed deep-learning model using a road distress dataset. This model can detect and classify road distress conditions in real-time, allowing self-driving cars to adjust their speed, trajectory, and other parameters based on the road conditions. This can improve the safety and efficiency of autonomous vehicles by reducing the risk of accidents and improving ride comfort.

The importance of such a dataset for self-driving cars lies in the fact that road distress conditions can affect the safety and performance of autonomous vehicles. For example, roughness can affect the stability and control of the vehicle, while cracking and rutting can cause vibrations and noise that affect the perception of the environment by the vehicle’s sensors. Faulting can also affect the vehicle’s trajectory and stability, especially at higher speeds.

To proceed with the experimentations, additional images (2973 in total) were taken from https://www.kaggle.com/datasets/shubhamadsul97/cdac-dai-pavement-distress-detection (accessed on 1 April 2023). These images contain different road conditions excluding potholes. The purpose of testing the model with two different datasets was to observe whether the distributed environment can adopt different kinds of input or not. The images were annotated in PASCAL format and divided into a 70% test set and a 30% validation set.

## 6. Experimental Settings

In this section, an in-depth experimental setting is applied to the proposed and traditional testbeds on the aforementioned dataset. Two testbeds are proposed as follows:

### 6.1. Proposed Low Resourceful Edge Devices Testbed

The proposed architecture contains combinations of edge devices to compete with powerful GPUs in terms of training time by utilizing different distributed techniques proposed in this research before. “Jetson Nano and Xavier both use ARM architecture” [34]. The ARM Cortex-A57 CPU is a high-performance 64-bit processor designed for energy-efficient computing. It is capable of executing multiple instructions in parallel, which helps to improve the overall performance of the system.

Similarly, a combination of Jetson Nano and Jetson Xavier is used in this research to form multi-dimensional edge clusters for distributed training of deep neural models with an extensive dataset to achieve accuracy in resource-constricted environments. Kubernetes is an open-source framework for automating containerized application deployment, scaling, and management, which serves the purpose of container orchestration in this study. Container orchestration and AI can be highly complementary technologies, as both require efficient and scalable management of distributed workloads. Container orchestration platforms, such as Kubernetes, provide a highly flexible and scalable infrastructure for deploying and managing containerized AI applications.

Edge clusters are developed using Kubernetes tokens to connect different edge devices in a parallel fashion, serving each as a GPU source to enable rigorous distributed training. Framework selection such as PyTorch, TensorFlow distributed API or hybrid pipeline depends on user requirements. A proposed environment for edge-distributed deep neural networks is shown in Figure 8 and Algorithm 2.
**Algorithm 2** Experimental Setup of Kubernetes Cluster with Edge Devices1:  **procedure** KubernetesSetup2:        Set up a Kubernetes master on a Jetson Nano & Jetson Xavier3:        Initialize the Kubernetes cluster on the master device4:        Retrieve the join token from the master device5:        **for** each Jetson Nano worker **do**6:              Join the Kubernetes cluster using the join token7:              Configure the worker device for Kubernetes8:              Apply required settings and resource limits9:        **end for**10:      Deploy the Distributed Pipeline via Jupyter Notebook Integration with edge devices11:      Set up monitoring tools for the cluster to measure computational cost12:      Verify that the cluster is running and all devices are connected13:  **end procedure**

### 6.2. Conventional Powerful Cloud GPU Testbed

The concept of the Google Cloud integrated GPU (graphics processing unit) is that graphics processing units (GPUs) can be added to the virtual machine (VM) instances using Compute Engine. These GPUs can speed up specialized workloads on virtual machines, such as machine learning and data processing. NVIDIA GPUs are provided in passthrough mode for the VMs, giving them direct control over the GPUs and their related memory.

The research conducted in this study utilized NVIDIA vGPU clusters consisting of Tesla K80 and P100 models. The Tesla K80 model can support up to eight GPUs with a maximum GPU memory of 96 GB GDDR5, as shown in Figure 9, while the P100 model can support up to four GPUs. To implement the virtualization environment powered by the NVIDIA virtual GPUs, the NVIDIA virtual GPU (vGPU) software is installed alongside the hypervisor at the virtualization layer. This software enables the creation of virtual GPUs, which allows each virtual machine (VM) to utilize the server’s hardware GPU. The software provides a graphics, or compute, driver for each VM, which enables the offloading of demanding operations to the hardware GPUs. This results in an improved user experience since the CPU is responsible for the bulk of the work. Furthermore, virtualized and cloud environments can support compute-intensive workloads, such as AI, data science, and demanding engineering and creative applications.

## 7. Results and Discussion

After performing the experimentation, the results were gathered and collected in several distributed parts for comparison as follows:

### 7.1. Computational Analysis

The PyTorch distributed API was compared with the TensorFlow API in a distributed fashion to analyze the image/second time and epoch time for the appropriate choice of API in the Google cluster environment. In Figure 10a,b the data set is categorized concerning the different batch sizes. In a nutshell, on four GPUs, the TensorFlow processing time was the least in comparison with PyTorch for data distribution. Furthermore, in image/second, the behavior of TensorFlow presented better results than PyTorch, except, at one point in two GPUs, TensorFlow surged in image/second due to communication overhead issues.

Moreover, the PyTorch distributed API was compared to the TensorFlow API in a distributed fashion to analyze the image/second time and epoch time for the appropriate choice of API in an edge cluster environment with up to three GPUs. Compared with the Google cluster for two initial GPUs, the data distribution takes added time for the TensorFlow libraries because of the communication overheads observed, as shown in Figure 11a,b.

The PyTorch distributed API was compared to the TensorFlow API in a distributed fashion to analyze the image/second time and epoch time for the appropriate choice of API in the Google cluster environment with up to four GPUs for model distribution. In the model distribution by the count of GPUs, the overall training time decreased with an increase in GPUs. There was a robust competition between TensorFlow and PyTorch for distributed model analysis, as the behavior was different for 1× the number of datasets and observed differently for 8×; overall, TensorFlow was fast because of the low communication connection, as shown in Figure 12a,b.

Compared with Google Colab, the overall time was increased in edge devices due to the connection time in and communication period. However, for resource-constricted environments with up to three GPUs, it was giving robust competition to the Google Cloud cluster, as shown in Figure 13a,b.

Moving toward hybrid pipeline analysis, a focused area of PyTorch, TensorFlow showed some limitations in this scenario. After the formation of the proposed hybrid testbed and experimentation, it was observed that the behaviors of PyTorch in the individual data or model distribution were different due to communication overheads. Nevertheless, it inculcates appropriate algorithms and NV-Links for solid communication. The hybrid pipeline significantly decreased the training time for the overall time (s) and image/seconds. The hybrid pipeline formed using the PyTorch library represented clear succession among TensorFlow hybridism in Google cluster analysis. Furthermore, the individual epoch was observed to be delayed because the initial two epochs are crucial for the formation of the linkage between gradients for forward and backward propagation. The results are analyzed and depicted in Figure 14a,b.

Similarly, the same hybridism was tested on edge devices. There was a slight increase in overall communication, but it was still remarkable for individual data and model-distributed training. Furthermore, this will be helpful in self-driving vehicles where resource-constricted ARM structures are observed in bulk. The results are depicted in Figure 15a,b.

The depth of losses is outside the scope of this study because this research focuses more on computational analysis. However, losses were observed for the model in two dimensions. Firstly, the log losses were calculated for detection analysis, including the ROI and actual value positions for the localization points. On the other hand, the cross-entropy loss is used to evaluate how well the model predicts the class of each object in an image detection scenario, as well as their location within the image.

The losses are observed in Figure 16 for the hybrid pipeline with respect to the epoch size. The overall and cross-entropy losses of the Google Colab cluster were reduced instantly due to strong cross-link GPUs and reduced communication overhead. However, the proposed hybrid technique showed remarkable competitiveness in loss tests.

### 7.2. Accuracy and Loss Analysis of Model

Training and validation accuracy are critical metrics in the context of distributed deep learning, where multiple nodes in a network are used to train a deep-learning model. In this scenario, it is important to measure the accuracy and validation of the model across all the nodes to ensure that the large model is performing well on a large and diverse dataset. In this section, the training and validation results of PyTorch, TensorFlow, and the proposed distributed deep-learning framework on an edge network are compared. As shown in Figure 17, it can be observed that the accuracy of PyTorch reached 95.1% and converged at epoch size 12, which is a competitive result considering the overhead and large distribution of the model.

Furthermore, the proposed strategy converged to an accuracy of 94.2% at epoch size 10, which is a reasonable result considering the overhead and the simultaneous processing of both the model and data with large parameters. These accuracy levels can be further improved by providing a solid communication platform to reduce overhead or by enhancing neural network parameters. However, this is not the aim of this study, as the focus of this study is mainly on distributed strategies to overcome computational complexity. If computational complexities are not addressed, they often lead to issues with the hardware of self-driving vehicles. The method proposed in this research is flexible and can be adopted in any challenging environment to achieve the best results.

In addition to these results, TensorFlow achieved an accuracy of 93.1%, which is reasonable but falls short compared to the other two methods. The major advantage of the proposed framework over the other two frameworks is that it processes both the model and data simultaneously, demonstrating that, even in larger scenarios, these accuracy levels will be solid. With a good communication strategy, the proposed framework has the potential to overcome many barriers.

In a distributed deep-learning system, the model is trained on multiple nodes, each with its own data and computation resources. The training and validation losses help to ensure that the model is learning and generalizing well across all the nodes. In a distributed deep-learning system, it is important to monitor the training and validation losses of each node, as well as the overall losses across all the nodes. If the losses are not consistent across all the nodes, this may indicate that the data distribution is not balanced or that some nodes are not contributing equally to the training process. As shown in Figure 18, we can observe the training and validation losses of PyTorch, TensorFlow, and the proposed distributed framework on the edge devices. With a distribution of 100 epochs, the average training losses of PyTorch are around 5–6%, the proposed framework shows losses of 6–7%, and TensorFlow shows losses of around 7–8%.

The proposed strategy has an advantage over the other two strategies as it processes both the data and the model in parallel, resulting in lower observed losses, compared to the other two frameworks that process either the data or the model in parallel and still face losses.

In summary, the proposed hybrid framework outperforms the state-of-the-art frameworks in terms of accuracy, losses, and distribution strategies. Its high level of accuracy, low observed losses, and efficient distribution strategies make it a safe and reliable choice for self-driving cars. The hybrid framework offers a compelling solution for autonomous vehicles that need to operate in complex and dynamic environments, ensuring their safe and efficient operation.

### 7.3. Detection Analysis of Pothole

After observing the training times, and comparing the clusters, the quality of the visual detection was observed specifically for the hybrid approach, which tends to be much more efficient for self-driving cars, as shown in Figure 19a,b. Overall, the quality for detection purposes was up to the mark.

### 7.4. Detection Analysis of Other Road Distress Conditions

Self-driving cars rely heavily on computer vision algorithms and sensors to detect and respond to real-time road conditions. From a computational perspective, detecting various road distress conditions, other than potholes, is crucial for ensuring autonomous vehicles’ safety, reliability, and efficiency. These road distress conditions, such as cracks, fissures, water pools, uneven surfaces, road leakages, and debris, can pose potential hazards to both self-driving cars and their passengers. To assess the effectiveness of our proposed methodology, we conducted experiments on different road distress conditions. The epoch size for training was set to 250, with a step size of 1563 for each epoch. The results showed that our proposed pipeline was efficient, as demonstrated by the bounding box quality and precision after the training on a dataset of road distress detection. As illustrated in Figure 20a,b, the results indicated that our proposed pipeline was up to the mark.

Additionally, a training and validation analysis was conducted to evaluate the robustness of the proposed methodology. As depicted in Figure 21a, the training and validation accuracies were observed to be 92%, which is competitive on a small dataset. Furthermore, as shown in Figure 21b, the training and validation losses were below 10%, indicating that the proposed methodology would be highly suitable for self-driving vehicles in various road situations when applied to a large-scale dataset.

## 8. Conclusions

This research addressed the challenge of pothole detection by autonomous vehicles due to inadequate training quality and embedded edge devices, arising from big data management and training difficulties. Furthermore, this study detected other road distress conditions to bolster its robustness. To enable comparison, two clusters were established, comprising an edge device cluster and a Google cluster. The PyTorch and TensorFlow distributed data and model APIs were assessed based on the total training duration and epoch size. A hybrid pipeline was suggested to overcome the limitations of individual distribution packages.

The hybrid pipeline, which integrates the all-reduce algorithms and NV-Links for forward and backward gradient propagation, demonstrated promising outcomes in terms of the total training time and epoch size, particularly within the PyTorch environment. In the Google cluster, the PyTorch-based hybrid pipeline, when processing 8× dataset repetitions on two GPUs, attained the shortest training time of 48 images/s; however, the first and second epoch sizes required additional time due to the gradient synchronization process. In the edge environment, these values were somewhat higher because of constraint limitations, but they still exhibited significant improvements when compared to separate distribution packages. The detection and cross-entropy losses were also evaluated to validate model accuracy on the given dataset. This work holds the potential to be extended to videographic analysis with high frame rates in smart cities for accurate, precise detection. Moreover, the validation and training losses were observed, and the overall model accuracy exceeded 92%, indicating a robust computer vision model.

## Figures and Tables

**Figure 1 sensors-23-04347-f001:**
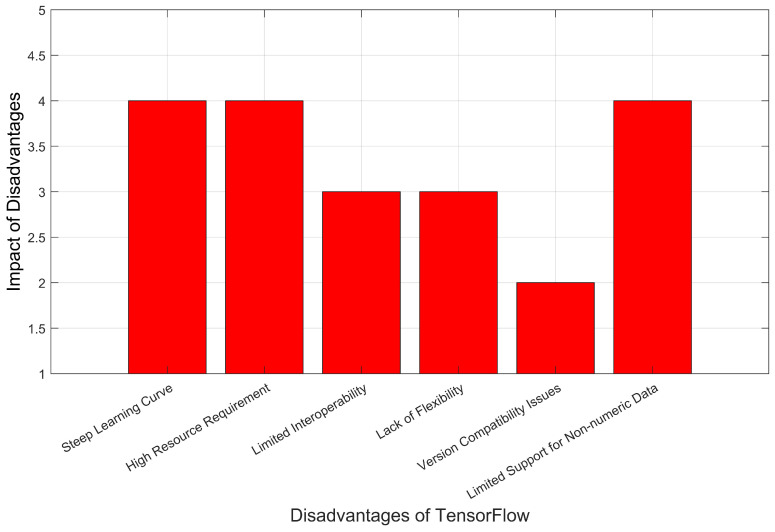
TensorFlow disadvantages.

**Figure 2 sensors-23-04347-f002:**
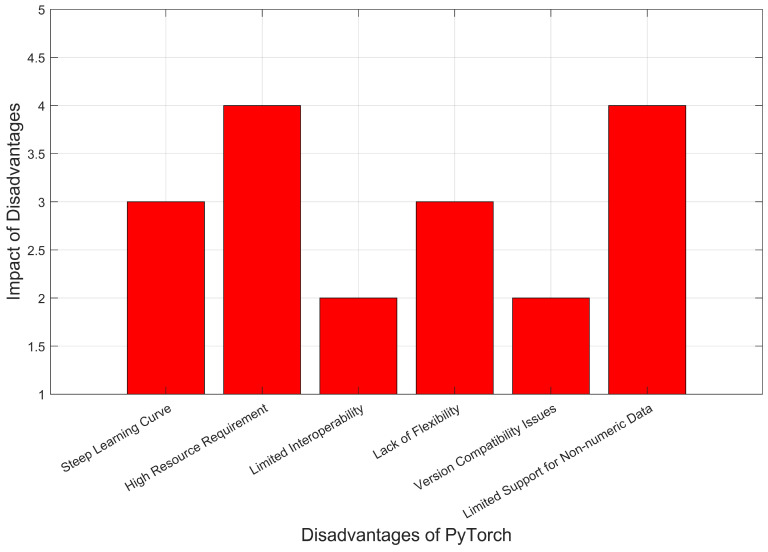
PyTorch disadvantages.

**Figure 3 sensors-23-04347-f003:**
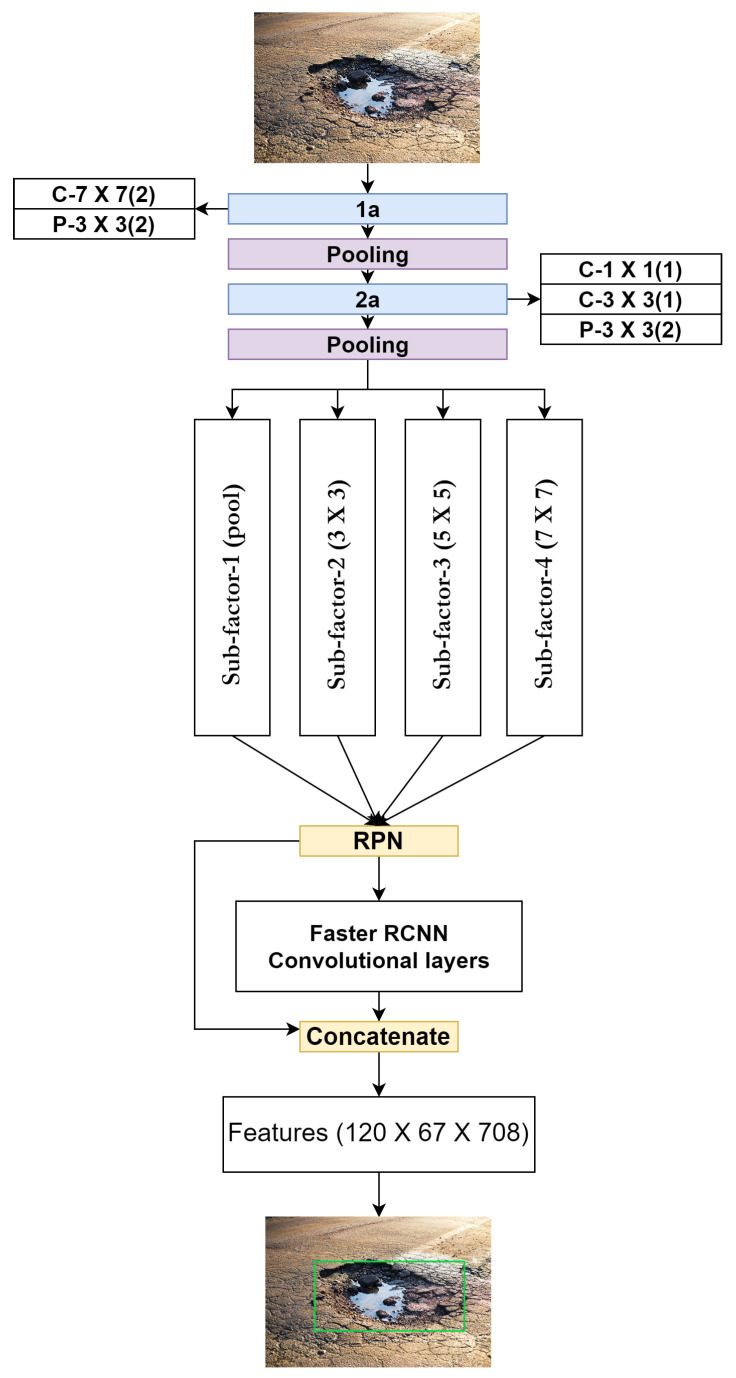
Distributed FactorNet model.

**Figure 4 sensors-23-04347-f004:**
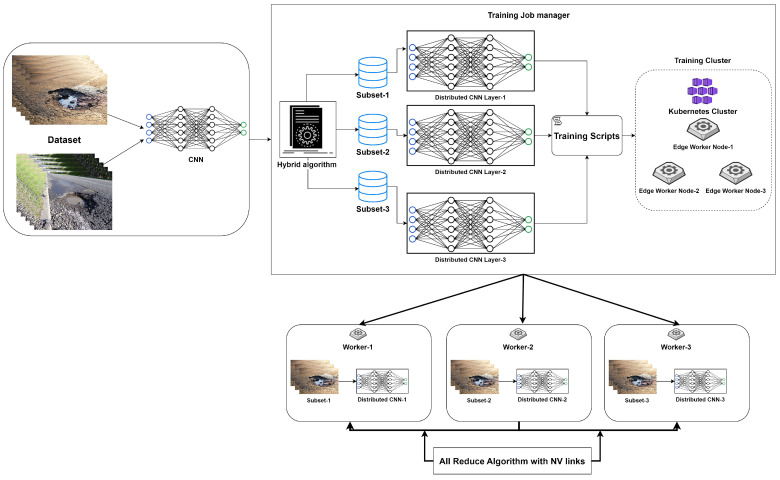
Conceptualization of proposed methodology.

**Figure 5 sensors-23-04347-f005:**
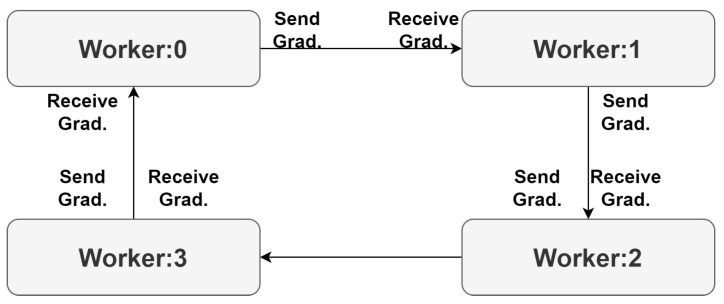
All-reduce exchange of distributed model parameters on distributed data.

**Figure 7 sensors-23-04347-f007:**
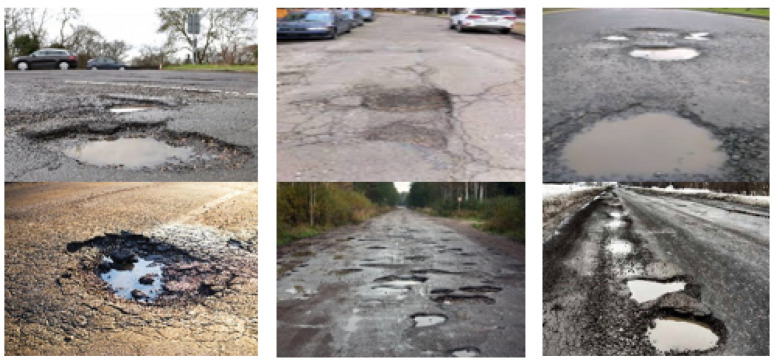
Pothole dataset for detection analysis.

**Figure 8 sensors-23-04347-f008:**
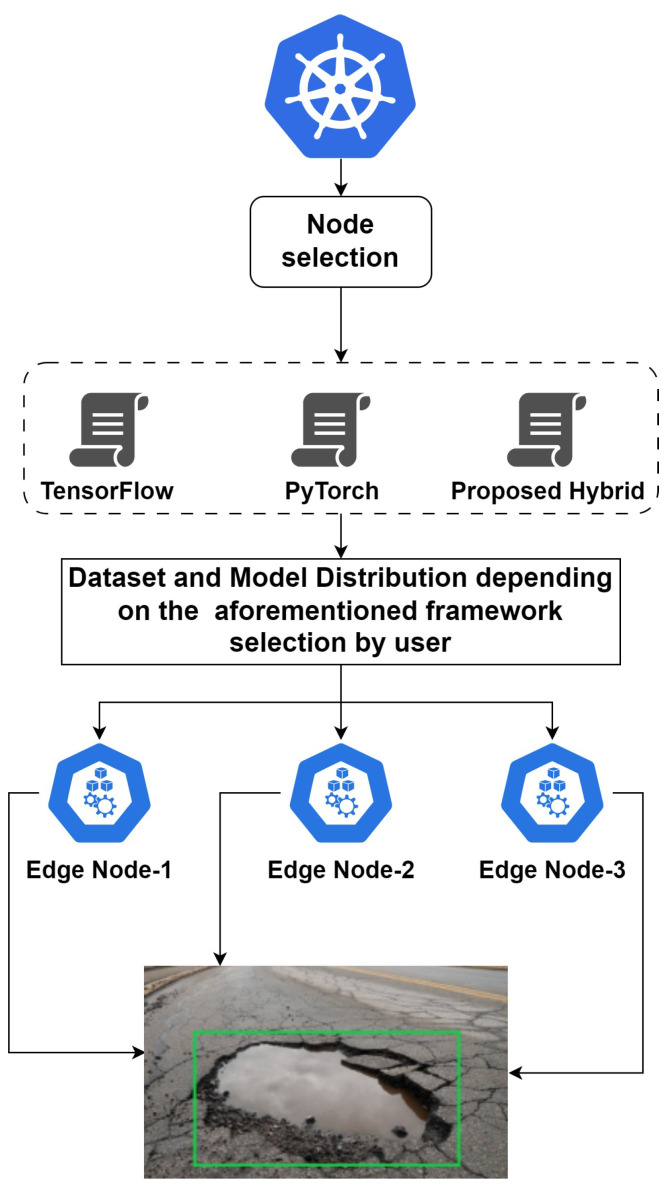
Proposed environment of edge distributed deep neural network for self-driving cars.

**Figure 9 sensors-23-04347-f009:**
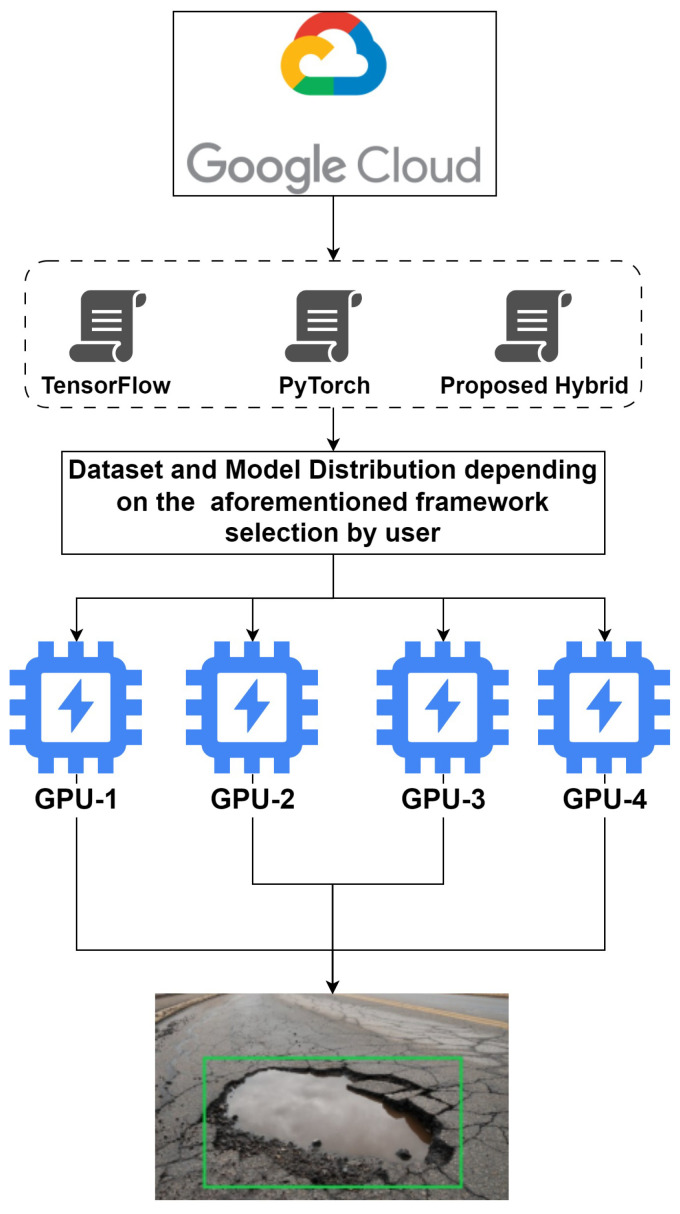
Conventional environment of GPU distributed deep neural network for self-driving cars.

**Figure 10 sensors-23-04347-f010:**
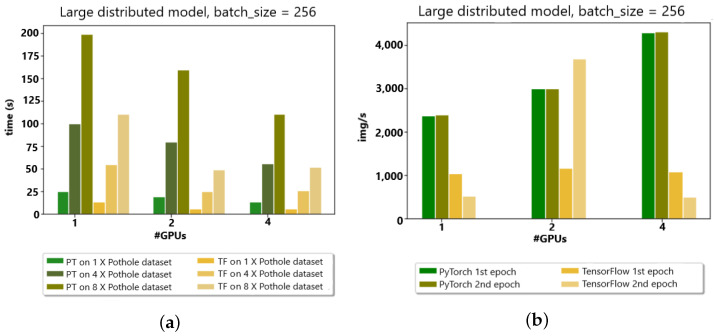
(**a**) Google Colab Cluster (4 GPUs) w.r.t. time (s). (**b**) Google Colab Cluster (4 GPUs) w.r.t. img/s.

**Figure 11 sensors-23-04347-f011:**
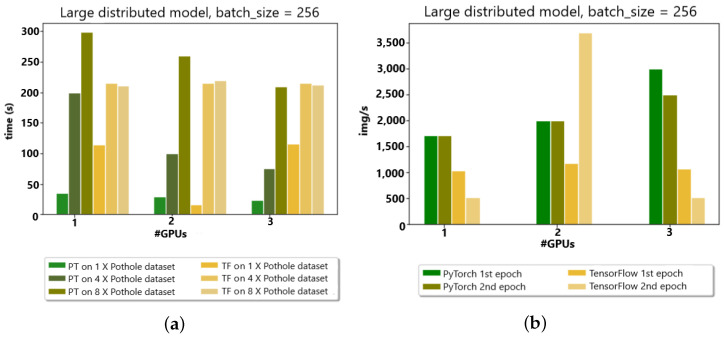
(**a**) Edge devices (3 GPUs) w.r.t. time (s). (**b**) Edge devices (3 GPUs) w.r.t. img/s.

**Figure 12 sensors-23-04347-f012:**
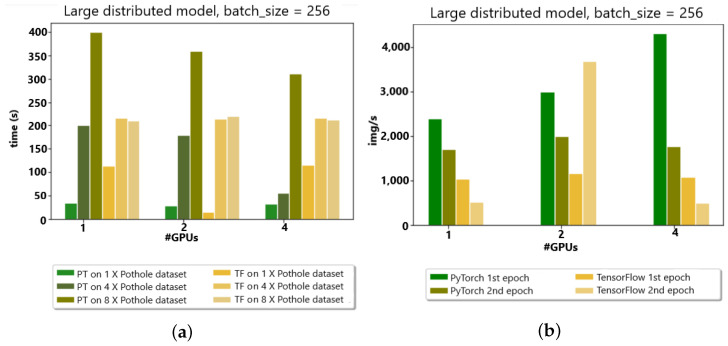
(**a**) Up to 4 GPUs w.r.t. time (second)). (**b**) Up to 4 GPUs w.r.t. img/s.

**Figure 13 sensors-23-04347-f013:**
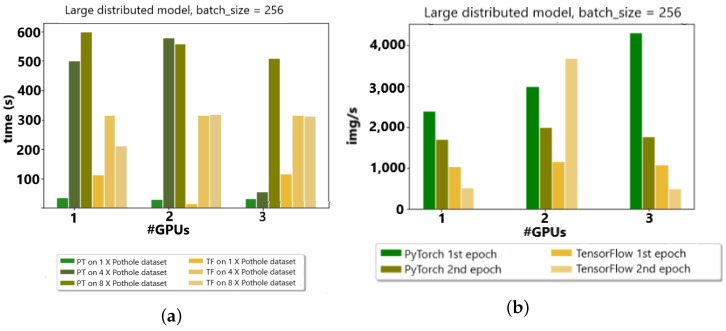
(**a**) Up to 3 GPUs w.r.t. time (second). (**b**) Up to 3 GPUs w.r.t. img/s.

**Figure 14 sensors-23-04347-f014:**
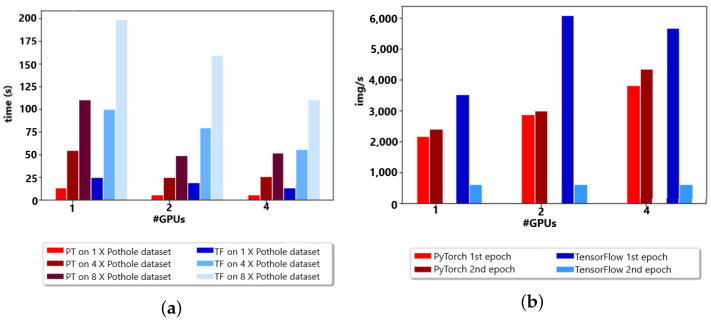
(**a**) Up to 4 GPUs w.r.t. time (seconds). (**b**) Up to 4 GPUs w.r.t. img/s.

**Figure 15 sensors-23-04347-f015:**
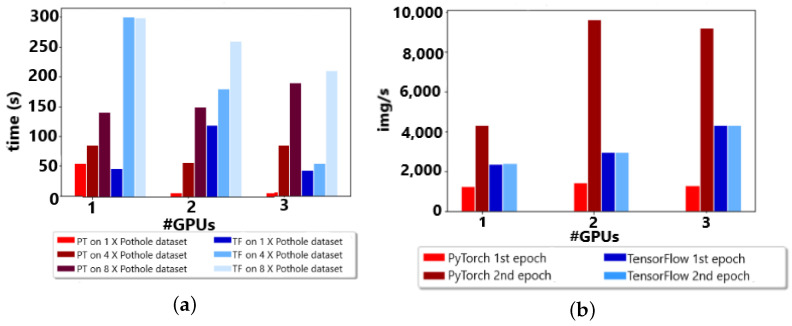
(**a**) Up to 3 GPUs w.r.t. time (second). (**b**) Up to 3 GPUs w.r.t. img/s.

**Figure 16 sensors-23-04347-f016:**
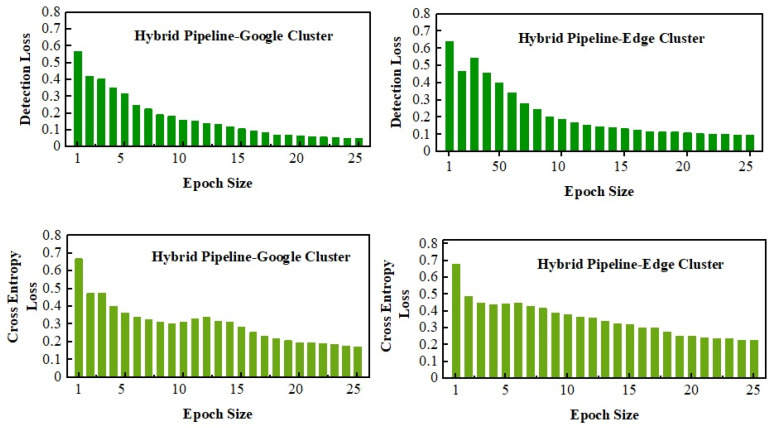
Validation analysis for hybrid pipeline.

**Figure 17 sensors-23-04347-f017:**
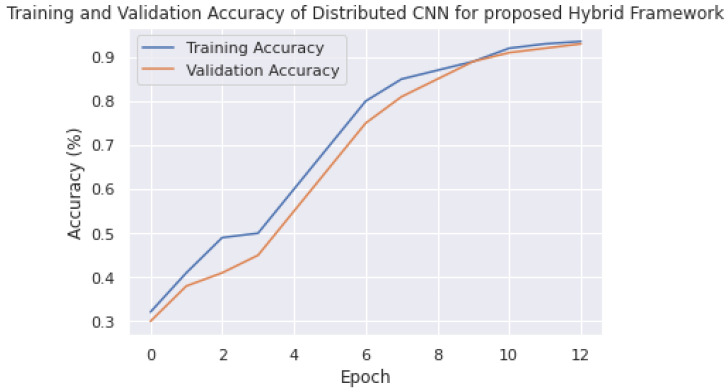
Training accuracy results of different frameworks on proposed edge testbed.

**Figure 18 sensors-23-04347-f018:**
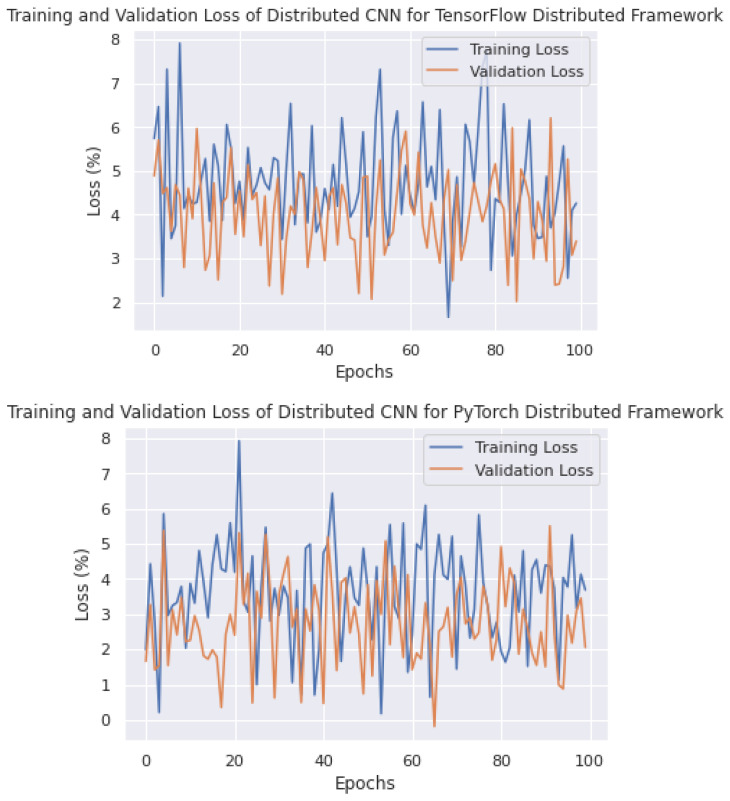
Training loss results of different frameworks on proposed edge testbed.

**Figure 19 sensors-23-04347-f019:**
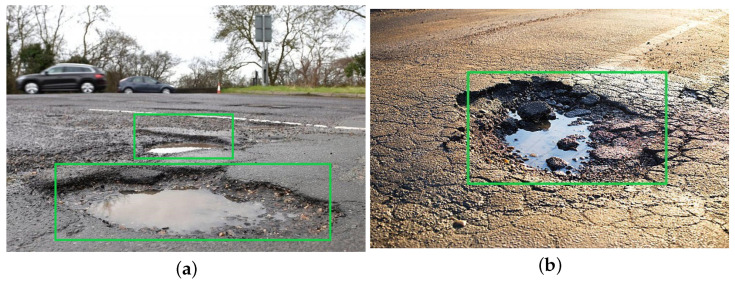
(**a**) Double pothole detection. (**b**) Single pothole detection.

**Figure 20 sensors-23-04347-f020:**
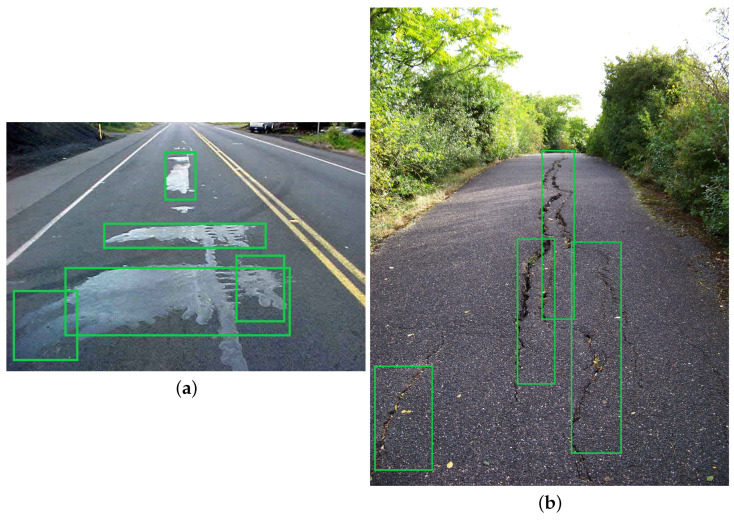
(**a**) Road leakages detection. (**b**) Road crack detection.

**Figure 21 sensors-23-04347-f021:**
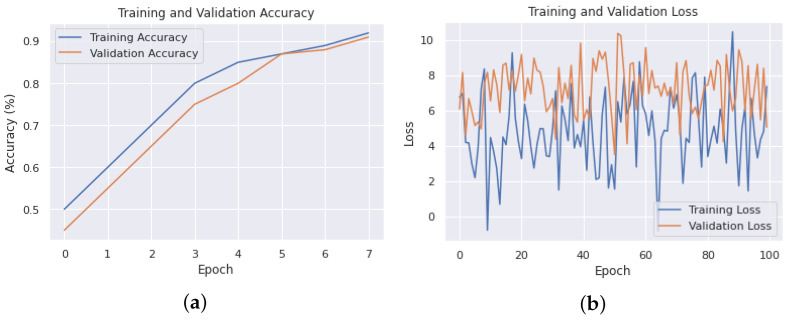
(**a**) Training and validation accuracies. (**b**) Training and validation losses.

**Table 1 sensors-23-04347-t001:** Comparative analysis of distributed PyTorch with distributed TensorFlow using key features.

Features	Distributed PyTorch	Distributed TensorFlow
Learning curve	Moderate	Steeper
Model deployment	Less mature deployment options	More mature deployment options (e.g., TensorFlow Serving)
Memory footprint (relative)	1.1	1
Mobile support	Less mature mobile support (PyTorch Mobile)	Mature mobile support (TensorFlow Lite)
Multi-node scaling efficiency (relative)	1	1.2
Static computation graphs	Not native, requires TorchScript	Native in TensorFlow 1.x, requires some adjustments in TensorFlow 2.x
CPU utilization (relative)	1	1.2
Performance	Slightly lower performance in some cases compared to TensorFlow	Slightly higher performance in some cases compared to PyTorch
ONNX support	Native support, but not all models are compatible	Requires third-party libraries (e.g., tf2onnx) for conversion
GPU utilization (relative)	1	1.2

## Data Availability

Data sharing not applicable.

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
