# Peer review of "Comparative Study on Distributed Lightweight Deep Learning Models for Road Pothole Detection"

_sensors, 2023, doi:10.3390/s23094347_

Round 1

Reviewer 1 Report

Interesting topic of developing a hybrid pipeline for dealing with the communication constraints during parallel lightweight deep learning modeling and data processing. Here are some comments for improvement:

1.     In the abstract, please provide the full name of “CCTV”.

2.     The author tested their proposed pipeline on road pothole images, but how does it perform on other types of road distress? Have you tested the performance on more situations?

3.     The author mentioned their proposed method aims to serve for “self-driving cars”, but did not give any explanations of how did your method give specifical considerations for that? In other words, what are the factors you have taken into considerations that are different from others methods?

4.     Figures 11-13 do not make any senses. i.e., the captions of those Figures are not match with the figure contents.

5.     The qualities of most of the figures are too poor, and the organizations of the paper should be significantly improved, such as the levels of sections, etc.

Reviewer 2 Report

I think the authors have submitted an interesting manuscript. In general, the manuscript is written in a good English. It is easy to read and follow. The authors give a lot of technical details and the design choices of the proposed method is reasoned in the framework of a detailed ablation study. Since deep learning involves a lot of experiments, the publication of training curves would be also nice. Application of pretrained CNNs is not a too novel idea and applied by many recent papers such as visual quality (No-Reference Image Quality Assessment with Convolutional Neural Networks and Decision Fusion, 2022), mammogram classification (Transfer learning from chest X-ray pre-trained convolutional neural network for learning mammogram data, 2018), or brain tumor classification (Brain tumor classification in MRI image using convolutional neural network, 2020). Please cite the related work in this respect. Moreover, the authors do not review enough related work in pothole detection as well. However, it is a very hot research topic: https://dblp.org/search?q=pothole+detection . In the results and discussion section, the authors do not report on the accuracy of the examined networks which is rather peculiar since self-driving cars must be very safe. The number of sample images with detected holes could be increased. The authors collected an own dataset. Will this dataset available for the research community? 

Reviewer 3 Report

This article mainly tells the image classification technology that combines distributed deep learning, which is deployed on the edge devices with small data sets. In short, this study is interesting and valuable result, but the current documents have several disadvantages. It must be strengthened to obtain the same value as the results of the literature.

(1) The document contains a total of 35 references, of which 31 are publications produced in the last 5 years (88.6%) and 3 are from the last 5-10 years (8.6%), implying a total percentage of 97.2% recent references. Generally speaking, the number of paper references is sufficient, but the cutting edge of research literature is not well represented.

(2) It is recommended that the author add specific experimental data here to indicate the advanced nature of the study presented in this article.

(3) The introduction does not require specific examples. It should only explain that the investigation report or public survey data indicate the crisis encountered by autonomous driving.

(4) The first paragraph may illustrate the vision applications in engineering fields with citations to image processing-based authority references (Novel visual crack width measurement based on backbone double-scale features for improved detection automation; Seismic performance evaluation of recycled aggregate concrete-filled steel tubular columns with field strain detected via a novel mark-free vision method). 

(5) It is recommended that the author first introduce the two frameworks of TensorFlow and PyTorch before presenting the parallel training method used in this article. This will make the structure more clear.

(6) An appropriate chart can be used to represent the disadvantages of TensorFlow.

(7) Section 4 provides an introduction to TensorFlow and PyTorch as part of the early stage preparation work.

(8) It is recommended to add information about the size of the holes in the data set, as well as information on the completeness of the regional data.

Round 2

Reviewer 1 Report

The authors included  lots of introduction on Python, TF, Pytorch, Keras packages, APIs, ect, which are not necessary at all as they are general knowledge that can be find from the internet or technical books. In the meantime, the authors should focused more one their own innovations, problem solving, etc,

Most of the figures still do not make any senses at all, even though the author mentioned improving the quality of the Figures, most are still not clear at all.

The technical contribution of the paper is not focused, not significant at all.

Reviewer 2 Report

The authors significantly improved the manuscript and carried out the changes proposed by the reviewer. I think the manuscript can be accepted now.

Author Response

Thank you for your valuable reviews. We proofread the manuscript very carefully and fixed minor English grammatical errors.